# Do Anxiety, Depression, Fear of Movement and Fear of Achilles Rupture Correlate with Achilles Tendinopathy Pain, Symptoms or Physical Function? [note 1]

**DOI:** 10.3390/jcm14020473

**Published:** 2025-01-13

**Authors:** George White, Fletcher Bright, Ebonie K. Rio, Ruth L. Chimenti, Myles C. Murphy

**Affiliations:** 1School of Health Sciences, The University of Notre Dame Australia, Fremantle, WA 6163, Australia; 2La Trobe Sport and Exercise Medicine Research Centre, La Trobe University, Bundoora, VIC 3083, Australia; 3Victorian Institute of Sport, Melbourne, VIC 3206, Australia; 4The Australian Ballet, Melbourne, VIC 3004, Australia; 5Department of Physical Therapy and Rehabilitation Science, University of Iowa, Iowa City, IA 52242, USA; 6Nutrition and Health Innovation Research Institute, School of Medical and Health Sciences, Edith Cowan University, Joondalup, WA 6027, Australia

**Keywords:** tendinopathy, psychological factors, TENDINS-A, Achilles, tendon

## Abstract

**Objectives:** To determine if psychological factors, such as anxiety, depression, fear of movement and fear of rupture are associated with increased tendon-related disability, quantified by the Tendinopathy Severity Assessment-Achilles (TENDINS-A). **Design:** Cross-sectional. **Setting:** Online Qualtrics survey. **Participants:** Sixty-eight participants (54% female) with Achilles tendinopathy and a mean (standard deviation) age of 40.1 (12.6) years. **Main Outcome Measures:** The TENDINS-A (including subscales of pain; symptoms such as stiffness; physical function), Patient Health Questionnaire-9, General Anxiety Disorder-7, Tampa Scale for Kinesiophobia and fear of tendon rupture. Associations were evaluated using generalised linear models (adjusting for age and sex), with significance accepted when *p* < 0.05. **Results:** Anxiety symptoms were positively associated with Achilles pain (*p* = 0.035), symptoms (*p* = 0.045) and physical function (*p* = 0.019). Depressive symptoms were negatively associated with symptoms (*p* = 0.045) but not pain (*p* = 0.078) or physical function (*p* = 0.429). Fear of movement was not associated with pain (*p* = 0.479), symptoms (*p* = 0.915) or physical function (*p* = 0.064). Fear of rupture was associated with pain (*p* = 0.042), but not symptoms (*p* = 0.797) or physical function (*p* = 0.509). **Conclusions:** Our research demonstrated anxiety symptoms and fear of rupture, not fear of movement or depressive symptoms, are positively associated with the severity of tendon-related disability. Therefore, clinicians should include an assessment anxiety symptoms and fear of rupture in their practice.

## 1. Introduction

Tendinopathy presents with focal, load-related tendon pain, decreased function and structural changes [1,2,3]. Achilles tendon pain occurs in either midportion or insertional regions [4] and improves over the course of activity, though pain and stiffness is typically worse 24 h following overload [5]. Pain location and pain behaviours are important, as they help delineate several differential diagnoses in the Achilles region [6].

Tendon pain is complex with features of nociceptive pain and some, but not all, of the central drivers seen in chronic pain [5]. Nociceptive pain occurs when specialised sensory neurons in the tissue (called nociceptors) become activated by mechanical, thermal or chemical changes in the tissue [5]. Alternatively, central sensitisation can include the increased sensitivity of neurons within the central nervous system [7]. The pain state appears to be maintained by complex mechanisms involving immune cells and processing changes within the spinal cord and brain [8].

Increased psychological distress may cause neurophysiological changes that enhance central sensitisation, peripheral sensitisation and alter motor function [5,6,7,8,9]. This highlights the potential relationship of psychological factors (anxiety, depression and fear of movement) with the severity of Achilles tendon pain and disability [10]. A systematic review by Stubbs et al. (2020) reported the association between psychological factors (anxiety, depression and kinesiophobia) and the severity of self-reported pain and disability across several different tendinopathy locations. They found a low certainty of evidence for a moderate association between depression, anxiety and kinesiophobia with greater self-reported pain and disability in rotator cuff tendinopathy and plantar heel pain [10]. They also reported a significant association between kinesiophobia and reduced function in Achilles tendinopathy [10]. However, there was limited research to establish a correlation between other psychological variables (anxiety and depression) and pain and disability in Achilles tendinopathy. This review reported low confidence in the estimates published, citing risk of bias, imprecision and indirectness in the included studies [10].

Most studies in the Achilles tendinopathy literature have relied on the Victorian Institute of Sport Assessment-Achilles (VISA-A) to assess the severity of Achilles tendinopathy-related disability. However, this outcome measure is now known to be invalid and should not be used [11]. It has recently been reported the severity of Achilles tendinopathy should only be measured using the new Tendinopathy Severity Assessment-Achilles (TENDINS-A) [12,13]. This measures the severity of Achilles tendinopathy across the domains of pain (e.g., warm up phenomena and increased pain with overload), symptoms (e.g., stiffness) and physical function (e.g., pain with loading). In addition, most studies assess kinesiophobia using the Tampa Scale for Kinesiophobia [14,15,16]. However, this outcome measure was initially designed and validated in a lower back pain population and has not been adapted to an Achilles tendinopathy population, so the questions included in the Tampa Scale for Kinesiophobia may not be relevant [17,18].

Several studies have examined the association between kinesiophobia and Achilles tendon pain and disability. As Achilles tendinopathy is a load-dependent condition, it is theorised that this may create a state of kinesiophobia, which could drive some tendinopathy pain and symptoms [19,20]. Furthermore, fear or movement, which we theorise includes fear of rupture, may present a barrier to participation in a rehabilitative exercise programme [17]. However, whilst psychological drivers of disability are well-documented in populations with chronic pain that are not necessarily load-related (e.g., osteoarthritis or lower back pain) [21,22,23,24], whether this relationship holds in a condition such as tendinopathy whereby constant pain is not a marker of the condition is of interest as it cannot be assumed to be present.

Chimenti et al. (2021) found a positive association between kinesiophobia and expected pain when asked to complete heel raises or hopping tasks [17]. In addition, participants with high kinesiophobia had greater pain at rest and stiffness. Alternatively, there was no association between severity of kinesiophobia and loading pain in a separate cross-sectional study [17]. However, the authors note potential self-selection bias with the loading pain data, since many of the participants refused to complete the hopping task. Murakawa et al. (2024) also found limited association between psychological factors (anxiety, depression, fear of movement and chronic pain self-efficacy) and Achilles tendon pain and loss of function [25]. However, the authors note a possible floor effect with the VISA-A since the sample population were mostly sedentary [25]. This research indicates that fear of movement may contribute to the pain mechanism seen in Achilles tendinopathy and may correlate with increased Achilles tendon-related disability, but it is not clear cut.

The present study aims to determine if psychological factors, such as anxiety, depression, fear of movement and fear of rupture, are associated with increased tendon-related disability, quantified by the TENDINS-A and measured across three subdomains: pain, symptoms and physical function. The secondary aims are to determine whether pain distribution and pain that increases with increasing tendon load (dose-dependent loading pain) relate to the severity of Achilles tendon pain, symptoms and physical function. Our additional exploratory aims were to determine the association between fear of movement and fear of rupture in Achilles tendinopathy.

We hypothesise the following: (i) higher anxiety, depression, fear of movement and fear of rupture are associated with higher severity of Achilles tendon-related disability across all three subdomains of pain, symptoms, and physical function; (ii) diffuse pain is associated with higher severity of Achilles tendon-related disability; (iii) non-dose-dependent loading pain is associated with lower Achilles tendon physical function, but not pain and symptoms; and (iv) higher fear of movement is associated with higher fear of tendon rupture.

## 2. Methods

### 2.1. Study Design

We conducted a cross-sectional study using online questionnaires via Qualtrics. Distribution of the initial questionnaire commenced from 1 December 2023. Recruitment ceased on 15 April 2024.

### 2.2. Recruitment

Participants were predominantly recruited using social media via global running Facebook^TM^ (Menlo Park, CA, USA) groups and allied health Instagram^TM^ (Menlo Park, CA, USA) accounts. A recruitment flyer was sent to physiotherapy and podiatry private practices across Australia and the survey was promoted at sports medicine lectures and other events hosted by the study authors.

### 2.3. Participants

Participants self-identified as having Achilles tendinopathy, which was confirmed via a correct location of pain using pain-mapping, as the TENDINS-A is not a diagnostic tool. This approach has been previously validated and provides near-perfect agreement with physician diagnosis [26], and we followed an identical procedure to our previous research [12,13,27].

Our sample size was dictated via a timeline constraint for recruitment as this research was conducted as a component of an integrated honours degree. Thus, recruitment was open for five months. A post hoc sensitivity analysis using G.Power 3.1.9.7 (α = 0.05, 1-β = 0.8, *n* = 68) reported we would be powered to detect effect sizes greater than 0.30.

### 2.4. Setting

Participants accessed the Qualtrics survey online and it was only available in English.

### 2.5. Variables

Participants completed several different outcome measures within a single Qualtrics questionnaire. The questionnaire included questions related to demographics, pain location, the TENDINS-A (subscales of pain, symptoms and physical function), depressive symptoms, anxiety symptoms, kinesiophobia, and fear of tendon rupture.

### 2.6. Demographics

Age (years), sex (male, female, intersex), gender (man, woman, non-binary, other), height (cm), weight (kg), race, country of residence, native language, additional languages spoken by the participant, ethnicity, education level, employment status and household income were documented (Appendix A). The broad reporting of demographic variables was designed to comply with international recommendations on the improved reporting of who is included in sports medicine and tendinopathy [28] research. This is important when many factors (e.g., ethnicity [29], sex [30] or socioeconomic status [31]) are poorly reported and/or often linked to treatment outcomes.

### 2.7. Pain Location

A standardised pain map allowed participants to select one of several different regions of posterior ankle pain (Appendix B) [6]. Pain location was classified as localised when pain locations A or D were selected. Pain location was classified as diffuse when pain locations B, C, E, F, G or H were selected.

### 2.8. Tendon-Related Disability

The Tendinopathy Severity-Achilles (TENDINS-A) was used to assess the severity of Achilles tendinopathy across three domains: pain, symptoms and physical function (www.tendinopathyseverityassessment.com, accessed on 10th January 2025). Each domain was scored as a percentage with 0% representing no disability and 100% representing complete disability. The TENDINS-A has sufficient content validity, structural validity, and test–retest reliability [12,13,32]. Further, the TENDINS-A does not exhibit item response bias for demographics such as age, sex, body mass index, education level, household income or physical activity level [32].

### 2.9. Load-Dependent Pain

Participants were classified as having load-dependent pain (Yes/No) based on their responses in the TENDINS-A, which is detailed in Appendix C. Load-dependent pain (Yes) was referred to as dose-dependent loading pain. Load-dependent pain (No) was referred to as non-dose-dependent loading pain.

### 2.10. Depressive Symptoms

The Patient Health Questionnaire-9 (PHQ-9) is a valid and reliable nine-item scale used to detect depressive symptoms [33]. The scale is scored between 0 and 27, with a higher score indicating greater depressive symptoms.

### 2.11. Anxiety Symptoms

The General Anxiety Disorder-7 (GAD-7) is a valid and reliable seven-item scale used to measure anxiety symptoms [34]. The scale is scored between 0 and 21, with a higher score indicating greater anxiety symptoms.

### 2.12. Kinesiophobia

The Tampa Scale for Kinesiophobia (TSK-17) is a 17-item questionnaire that assess fear of movement [35] that was designed for people with lower back pain. The scale is scored between 17 and 68, with a total score greater than 34 indicating kinesiophobia. The Tampa Scale of Kinesiophobia was modified by consensus of the research team to include Achilles tendinopathy-specific language (Appendix D). This modification aimed to improve the comprehensibility of the TSK-17, which has not been previously performed in Achilles tendinopathy [36], based on research in other pathologies whereby the language of the TSK was modified, including pathologies such as cancer survivors [37], patellofemoral pain [38], shoulder instability [39] and temporomandibular pain [40].

### 2.13. Fear of Tendon Rupture

We included a single item to assess the fear of tendon rupture, based on the Tampa Scale for Kinesiophobia structure, which was “Due to my tendon pain, the tendon is likely to snap/rupture”. Participants were asked to respond using a four-point Likert scale (0 = strongly disagree, 1 = disagree, 2 = agree, 3 = strongly agree).

## 3. Data Analysis

### 3.1. Data Management

Data were downloaded from Qualtrics as a Microsoft Excel spreadsheet and the data were visually inspected and transformed to be compatible with SPSS Statistics. Participants who did not complete the TENDINS-A were excluded from the data analysis. Where participants did not complete a question on an outcome measure, they were assigned the worst possible score for that item (<5% of participants). Pain locations were recategorized when the participant selected multiple regions that overlapped. Specifically, if C and B were both reported then the pain region was assigned B; if D and B were both reported then the pain region was assigned B; and if F and B were both reported the pain region was assigned B.

### 3.2. Statistical Analysis

Following the data transformation described above, the Microsoft Excel spreadsheet was imported into SPSS (Version 28.0.1.0). Descriptive statistics were performed to determine variable count, mean, standard deviation (SD), median, range and distribution, as indicated. The mean, SD, median and range of the total score for TENDINS-A subsets (Pain, Symptoms and Physical Function), PHQ-9, GAD-7, Tampa Scale for Kinesiophobia and fear of rupture were calculated and assessed for normality.

The associations between anxiety, depression, fear of movement or fear of rupture with the severity of each disability domain (pain, symptoms, physical function) were explored using generalised linear models, adjusted for participant age and sex. Beta-estimates and 95% confidence intervals are presented, with the significance set as *p* < 0.05. The following assumptions for generalised linear models were checked to ensure that modelling was appropriate: data were normally distributed, the x–y relationship was linear, residuals were checked and did not have a relationship to the predicted values and there was no substantive collinearity between variables.

Binary logistic regression was used to explore if localised pain (Yes/No) or load dependent pain (Yes/No) were associated with the TENDINS-A (Pain), TENDINS-A (Symptoms) or TENDINS-A (Function). Beta-coefficients and 95% confidence intervals are presented, with the significance set as *p* < 0.05. The following assumptions for binary logistic regression were checked to ensure that modelling was appropriate: data were normally distributed, observations were independent of each other and there was no substantive collinearity between variables.

The association between fear of movement and fear of tendon rupture was calculated using a non-parametric spearman’s rho test with 95% confidence intervals and the significance set at *p* < 0.05.

## 4. Results

A total of 86 participants attempted the survey. A total of 18 participants were excluded from the study due to incomplete data on the TENDINS-A. We included 68 participants in analysis (54% female sex) who were of predominantly ‘Australian’ ethnicity (*n* = 34, 50%) and performed physical activity most days (*n* = 58, 85%). Participants were otherwise from a diverse educational and socioeconomic background (Appendix A).

### 4.1. Baseline Characteristics

Participants were a mean (SD) age of 40.1 (12.6) years with a BMI of 25.0 (4.5) kg/m^2^. The baseline characteristics for tendon-related disability, depressive symptoms, anxiety symptoms, kinesiophobia and fear of rupture are located in Table 1.

### 4.2. Pain Locations

The three most common locations (count) for pain with loading were the insertional Achilles (n = 11), peritendon (n = 19) and the midportion Achilles (n = 12). A total of 15 participants reported diffuse pain in the Achilles tendon region (i.e., reported both mid-portion and insertional Achilles tendinopathy).

### 4.3. Load-Dependent Pain

38 participants were classified as having dose-dependent loading pain (load-dependent pain (Yes)) and 30 participants were classified as having non-dose-dependent loading pain (load-dependent pain (No)).

## 5. Association Between TENDINS-A (Pain) and Psychological Factors

A significant, positive association was detected between anxiety symptoms and TENDINS-A (Pain), whereby for every 1 point higher on the GAD-7 the TENDINS-A (Pain) score was 1.9% higher (*p* = 0.035). A significant, positive association was detected between fear of tendon rupture and pain, whereby for every 1 point higher for the fear of rupture the TENDINS-A (Pain) score was 12.4% higher (*p* = 0.042). There was no association between depressive symptoms and TENDINS-A (Pain) or kinesiophobia and TENDINS-A (Pain), with the complete analysis presented in Table 2.

## 6. Association Between TENDINS-A (Symptoms) and Psychological Factors

A significant, positive association was detected between anxiety symptoms and TENDINS-A (Symptoms), whereby for every 1 point higher on the GAD-7 the TENDINS-A (Symptoms) score was 1.7% higher (*p* = 0.045). A significant, negative association was detected between depression symptoms and TENDINS-A (Symptoms), whereby for every 1 point higher on the PHQ-9 the TENDINS-A (Symptoms) score was 1.6% lower (*p* = 0.045). There was no association between kinesiophobia and TENDINS-A (Symptoms) or fear of rupture and TENDINS-A (Symptoms), with the complete analysis presented in Table 3.

## 7. Association Between TENDINS-A (Physical Function) and Psychological Factors

A significant, positive association was detected between anxiety symptoms and TENDINS-A (Physical Function), whereby for every 1 point higher on the GAD-7 the TENDINS-A (Physical Function) score was 2% higher (*p* = 0.019). A significant association was detected between sex and TENDINS-A (Physical Function), whereby females had a 13.4% higher TENDINS-A (Physical Function) score (*p* = 0.016). There was no association between depression symptoms and TENDINS-A (Physical Function), kinesiophobia and TENDINS-A (Physical Function) or fear of rupture and TENDINS-A (Physical Function), with the complete analysis presented in Table 4.

## 8. Tendinopathy-Related Disability in Diffuse Versus Localised Pain

The distribution of tendon pain (either being local or diffuse) was not associated with TENDINS-A (Pain) score (*p* = 0.190), TENDINS-A (Symptoms) score (*p* = 0.532) or TENDINS-A (Physical Function) score (*p* = 0.146). The complete analysis is presented in Table 5.

## 9. Tendinopathy-Related Disability in Dose-Dependent and Non-Dose-Dependent Loading Pain

Having load-dependent pain was associated with a better (lower) TENDINS-A (Physical Function) score (*p* = 0.016). However, there was no association between having load-dependent pain and participants’ TENDINS-A (Pain) score (*p* = 0.444) or TENDINS-A (Symptoms) score (*p* = 0.757). The complete analysis is presented in Table 6.

## 10. Association Between Fear of Movement and Fear of Rupture

There was a moderate significant positive relationship between fear of movement and fear of rupture (rho = 0.563, 95% CI = 0.336 to 0.726, *p* < 0.001).

## 11. Discussion

The primary aims were to determine if psychological factors were associated with increased tendon-related disability, quantified by the TENDINS-A. As hypothesised, our research demonstrated significant positive associations between anxiety symptoms and fear of rupture with TENDINS-A disability. However, contrary to our hypothesis, kinesiophobia was not associated with any TENDINS-A subdomains and depression symptoms were negatively associated with TENDINS-A (Symptoms). Individuals with dose-dependent loading pain had better physical function compared to individuals with non-dose-dependent loading pain. Surprisingly, compared to localised pain, diffuse pain was not significantly associated with increased Achilles tendon-related pain, stiffness and loss of physical function. This is the first study to establish the relationship between psychological variables and Achilles tendon-related disability using a validated outcome measure (TENDINS-A).

The current study found that higher levels of anxiety were specifically associated with increased TENDINS-A (Pain), TENDINS-A (Symptoms) and TENDINS-A (Physical Function). Thus, our results for anxiety symptoms are contrary to Murakawa et al. (2024), who found limited associations between anxiety and tendon-related disability [25]. This inconsistency may be due to several reasons. Firstly, Murakawa et al. used the VISA-A to measure the severity of Achilles tendinopathy. As the VISA-A has insufficient content and structural validity, it does not accurately measure the severity of Achilles tendinopathy [11]. Additionally, the authors note a possible ceiling effect with the VISA-A since the sample population were mostly sedentary [25]. Such limitations may limit the capability of detecting an association between psychological factors and the severity of Achilles tendinopathy. Our study is the first to examine the association between anxiety or depressive symptoms and the severity of Achilles tendon pain, stiffness and loss of physical function using a validated outcome measure (TENDINS-A).

The mean fear of movement (kinesiophobia) score of participants in our study suggested participants in our cohort did not display, at a group level, high levels of kinesiophobia (scores above 37 indicate kinesiophobia). However, the range of scores recorded by participants was large, and some included participants had severe kinesiophobia [17]. Our study found no association between kinesiophobia and Achilles tendon pain, stiffness or physical function. This outcome somewhat aligns with the research by Chimenti et al. (2021), which found increased fear of movement was associated with greater expected pain, pain at rest and stiffness, but not increased pain with loading [17]. In this study, the authors reported self-selection bias since the participants were able to elect to avoid tasks, including heel raises and hops, that they deemed to be too provocative. Therefore, the analysis from this study may have under-reported the association between kinesiophobia and pain with loading. Nevertheless, the results were comparable to our current findings. Our study also used the TSK-17 rather than the TSK-11, as used in the study by Chimenti et al., which seems to indicate that, irrespective of the scale, kinesiophobia is not an important contributor to Achilles tendinopathy-related dysfunction. The current study is the first to adapt the language of the Tampa Scale of Kinesiophobia to include replace the word ‘injury’ with ‘Achilles tendon’, which we hypothesise may improve comprehensibility. However, future research could look to formally revise the TSK-17 for Achilles tendinopathy using established procedures from other conditions [39] or outcomes.

Increased fear of rupture was associated with increased tendon pain, but not stiffness of physical function. Despite measuring a similar construct to the TSK-17, it may be that in a tendinopathy population, asking a single question related to rupture is more informative than the larger TSK-11 or TSK-17 outcome measure. To our knowledge, this is the first study to examine the association between fear of tendon rupture and the severity of Achilles tendon disability. This is important since a fear of tendon rupture may present a barrier to exercise participation. Further research is needed to determine the impact this fear has on rehabilitation.

Our study found no association between diffuse pain (that may be indicative of central sensitisation) and increased Achilles tendon pain, stiffness, and loss of physical function. A sample size of n = 68 may have been underpowered to identify an association between these variables; however, inspection of the effect sizes does not appear to indicate a meaningful association.

Participants with non-dose-dependent loading pain experienced more severe pain with loading compared to individuals with dose-dependent loading pain. Interestingly, generic pain symptoms and stiffness were not significantly associated with loading pain profiles. A recent study by Murphy et al. (2024) found that questions relating to pain with loading provided high clinical relevance when discriminating between mild and severe Achilles tendinopathy [27]. Conversely, questions relating to stiffness offered less discriminatory capacity [27]. However, Murphy et al. (2024) does not specify whether a dose response to loading was significant in distinguishing between mild and severe disability. Thus, our findings answer this question and suggest individuals who experience tendon pain independent of tendon load have more severe Achilles tendon-related disability.

In the context of the broader musculoskeletal pain literature, this study reports some contrasting findings. We did not find that increased depressive symptoms were positively associated with pain, which has been reported in other musculoskeletal pain conditions with massive healthcare burdens such as rotator cuff tendinopathy [41,42], lower back pain [43], chronic neck pain [44] and osteoarthritis [45]. The reason for this may be people with Achilles tendinopathy are generally physically active (e.g., 85% of our cohort self-reported performing moderate to vigorous physical activity most days), which counteracts depressive symptoms. We also did not detect any associations between fear of movement and Achilles tendon pain, symptoms or function. Fear of movement has been associated with symptom severity in rotator cuff tendinopathy [41,46], chronic lower-back pain [47,48], chronic neck pain [49,50] and osteoarthritis [51,52]. This difference may be due to the intermittent nature of tendon pain, which has a clear mechanical nature [5].

Finally, longitudinal studies with larger samples that can explore more than just aetiology are needed. Specifically, studies that can evaluate the moderating and mediating role that psychosocial factors [53,54] have on tendinopathy, as per other conditions [55], would be important to progress clinical practice. Further, the role of other psychological variables beyond anxiety, depression and fear of movement or rupture may be important. Psychological constructs that have been proposed to be important include the fear of pain [56,57] and pain catastrophizing [58,59,60], which may provide novel insights into the management of Achilles tendinopathy.

## 12. Clinical Implications

This study highlights the following important clinical implications: (i) it is important to consider the possibility of an elevated fear of tendon rupture, and not fear of movement, when managing a patient with Achilles tendinopathy; (ii) elevated anxiety symptoms are associated with the severity of pain, symptoms and physical function in Achilles tendinopathy and should be screened as a routine part of clinical assessment; and (iii) establishing whether a patient has increased tendon pain with increased tendon load should be included in the routine clinical assessment of Achilles tendinopathy, since this appears to be directly associated with pain severity. Further research is warranted to explore the longitudinal impact these psychological factors have on recovery using the TENDINS-A to measure changes in the severity of disability.

## 13. Limitations

A sample size of *n* = 68 was sufficient to perform the statistical analysis required for this study’s primary aims. The participants in this study had low levels of anxiety and depressive symptoms and while this is consistent with other studies in the Achilles tendinopathy studies the findings may not be generalisable to people with severe anxiety or depressive symptoms. Our study only explored associations, so we are unable to make any inference in relation to causality. The population in our study was of predominantly Australian ethnicity, which may limit the generalisability of our results to other countries or ethnicities. Our recruitment method included the dissemination of our research flyer via social media, which may predispose this study to sampling bias towards participants with more disability or more severe psychological factors. However, given the large distribution of TENDINS-A scores and the relatively low prevalence of severe psychological factors, our data would suggest this is unlikely.

## 14. Conclusions

Our research demonstrated significant correlations between anxiety and depression with the severity of tendon-related disability. Fear of movement was only associated with physical function, whereas fear of rupture was associated with pain and physical function.

## Figures and Tables

**Table 1 jcm-14-00473-t001:** Participant baseline characteristics.

Variables	Mean	SD	Median	Range
TENDINS-A (Pain) ^	51.47	26.39	52.27	100.00
TENDINS-A (Symptoms)	54.60	24.14	50.00	87.50
TENDINS-A (Physical Function)	45.29	31.50	43.33	100.00
GAD-7 Raw Score/21 (Anxiety) ^	6.74	7.71	4.00	21.00
PHQ-9 Raw Score/27 (Depression) ^	7.81	9.84	3.00	27.00
Tampa Scale of Kinesiophobia Raw Score/68 (Fear of Movement)	42.40	14.52	37.00	51.00
Fear of Rupture Raw Score/3 ^	2.13	0.73	2.00	3.00

Legend: SD = Standard Deviation; TENDINS = Tendinopathy Severity Assessment—Achilles; GAD-7 = Generalised Anxiety Disorder—7; PHQ-9 = Patient Health Questionnaire—9; ^ indicates data are nonparametric.

**Table 2 jcm-14-00473-t002:** Generalised linear model evaluated the association between TENDINS-A (Pain) and psychological factors.

Variable	Beta-Estimate	95% Confidence Interval: Lower Limit	95% Confidence Interval: Upper Limit	*p*-Value
GAD-7	1.922	0.132	3.712	0.035
PHQ-9	−1.513	−3.196	0.171	0.078
Tampa Scale of Kinesiophobia	−0.293	−1.104	0.518	0.479
Fear of Rupture	12.438	0.462	24.413	0.042
Sex (Female)	2.737	−9.129	14.602	0.651
Age	0.298	−0.164	0.760	0.206
Intercept	22.781	−7.957	53.520	0.146

Legend:GAD-7 = Generalised Anxiety Disorder—7; PHQ-9 = Patient Health Questionnaire—9. Akaike’s Information Criterion (AIC) = 642.

**Table 3 jcm-14-00473-t003:** Generalised linear model evaluated the association between TENDINS-A (Symptoms) and psychological factors.

Variable	Beta-Estimate	95% Confidence Interval: Lower Limit	95% Confidence Interval: Upper Limit	*p*-Value
GAD-7	1.690	0.036	3.344	0.045
PHQ-9	−1.594	−3.149	−0.038	0.045
Tampa Scale of Kinesiophobia	−0.041	−0.790	0.708	0.915
Fear of Rupture	1.454	−9.609	12.517	0.797
Sex (Female)	8.263	−2.698	19.224	0.140
Age	0.097	−0.330	0.523	0.657
Intercept	45.919	17.523	74.315	0.002

Legend:GAD-7 = Generalised Anxiety Disorder—7, PHQ-9 = Patient Health Questionnaire—9. Akaike’s Information Criterion (AIC) = 631.

**Table 4 jcm-14-00473-t004:** Generalised linear model evaluated the association between TENDINS-A (Physical Function) and psychological factors.

Variable	Beta-Estimate	95% Confidence Interval: Lower Limit	95% Confidence Interval: Upper Limit	*p*-Value
GAD-7	1.982	0.323	3.641	0.019
PHQ-9	−0.630	−2.190	0.930	0.429
Tampa Scale of Kinesiophobia	0.709	−0.042	1.460	0.064
Fear of Rupture	3.741	−7.355	14.837	0.509
Sex (Female)	13.470	2.476	24.463	0.016
Age	0.321	−0.107	0.749	0.142
Intercept	−21.385	−49.866	7.095	0.141

Legend: GAD-7 = Generalised Anxiety Disorder—7; PHQ-9 = Patient Health Questionnaire—9. Akaike’s Information Criterion (AIC) = 632.

**Table 5 jcm-14-00473-t005:** Binary logistic regression evaluating association of tendinopathy-related disability with localised tendon pain.

Variable	Beta-Coefficient	95% Confidence Interval: Lower Limit	95% Confidence Interval: Upper Limit	*p*-Value
TENDINS-A (Pain)	0.986	0.965	1.007	0.190
TENDINS-A (Symptoms)	1.007	0.984	1.031	0.532
TENDINS-A (Physical Function)	0.987	0.969	1.005	0.146
Constant	1.223			0.800

Legend: TENDINS-A = Tendinopathy Severity Assessment-Achilles; Nagelkerke R Square = 0.086.

**Table 6 jcm-14-00473-t006:** Binary logistic regression evaluating association of tendinopathy-related disability with having load-dependent tendon pain.

Variable	Beta-Coefficient	95% Confidence Interval: Lower Limit	95% Confidence Interval: Upper Limit	*p*-Value
TENDINS-A (Pain)	0.992	0.971	1.013	0.444
TENDINS-A (Symptoms)	1.004	0.981	1.026	0.757
TENDINS-A (Physical Function)	0.979	0.963	0.996	0.016
Constant	4.791			0.060

Legend: TENDINS-A = Tendinopathy Severity Assessment-Achilles; Nagelkerke R Square = 0.140.

## Data Availability

De-identified data for this research project are available from the corresponding author upon reasonable request.

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
