# Peer review of "Do Anxiety, Depression, Fear of Movement and Fear of Achilles Rupture Correlate with Achilles Tendinopathy Pain, Symptoms or Physical Function?†"

_jcm, 2025, doi:10.3390/jcm14020473_

Round 1

Reviewer 1 Report

Comments and Suggestions for Authors

Well done on an important article. The only edit I would suggest is for another proofread (very very minor) for better flow. Other than that, it looks ready. 

Author Response

Well done on an important article. The only edit I would suggest is for another proofread (very very minor) for better flow. Other than that, it looks ready. 

    1. Thank you for your comments we have provided a proof read of the document and removed some sub-headings to assist with flow.

Reviewer 2 Report

Comments and Suggestions for Authors

----------------------General comments

The aim of this study is to determine if psychological factors are associated with increased tendon-related disability, as measured by the TENDINS-A. The article’s structure, particularly in the Methods and Results sections, falls significantly short of the standards expected. The Introduction outlines a rationale and gaps that are not addressed by the chosen methods. The sample size is small and unexplained, and statistical power is absent. Additionally, the study fails to use multiple correlation models or complex networks, which could provide new insights in the studied context.

----------------------Specific comments

----------Introduction

-Avoid excessive use of the active voice (e.g., "They found...").

-The introduction would benefit if the penultimate and third-to-last paragraphs were switched. The close of the third-to-last paragraph introduces limitations that appear to be the focus of the article.

-After introducing an acronym, ensure consistent use (e.g., VISA-A).

-The last paragraph needs improvement. Critical points should appear earlier. Instead of a separate section for the aim, consider integrating it into the last paragraph to present the objective and hypotheses concisely.

----------Methods

-The sample size is exceptionally low for a study conducted with an online questionnaire.

-Inclusion and exclusion criteria were not presented.

-The approach used for identifying Achilles tendinopathy lacks adequate detail and needs expansion.

-Information regarding validation and internal/external consistency was not provided for the questionnaires.

-Statistical Analysis: This is the primary concern. As highlighted in the introduction, various studies have explored these variables in isolation. This study would benefit from innovative insights, which could be achieved through multiple correlations or complex network models.

----------Results

-This section lacks clarity and should not be repeatedly divided into subsections. The format resembles a report rather than a research article.

Author Response

  1. Avoid excessive use of the active voice (e.g., "They found...").
    1. Thank you for this comment and we appreciate different academics have strong preferences on use of the active versus passive voice. The preference of all study authors is the active voice, and we would prefer to not change this being our preference. However, if the journal only accepts a passive voice, we will amend this.
  2. The introduction would benefit if the penultimate and third-to-last paragraphs were switched. The close of the third-to-last paragraph introduces limitations that appear to be the focus of the article.
    1. We have amended the order of the paragraphs as suggested.
  3. After introducing an acronym, ensure consistent use (e.g., VISA-A).
    1. Apologies for this oversight it has been amended.
  4. The last paragraph needs improvement. Critical points should appear earlier. Instead of a separate section for the aim, consider integrating it into the last paragraph to present the objective and hypotheses concisely.
    1. We have amended the introduction to address your concern.
  5. The sample size is exceptionally low for a study conducted with an online questionnaire.
    1. Thank you for your concern, we have provided further detail to our sample size section to address this. It is also important to note that as we detected significant results, we are not at risk of type II error for our primary outcomes based on our current sample size.
  6. Inclusion and exclusion criteria were not presented.
    1. Our inclusion criteria are presented in the participants section, with participants being included if they had Achilles tendon pain and dysfunction in the correct region of the tendon, as identified via a pain map.
  7. The approach used for identifying Achilles tendinopathy lacks adequate detail and needs expansion.
    1. Thank you we have provided some further detail here, and there is additional detail in the appendix section to assist the reviewer.
  8. Information regarding validation and internal/external consistency was not provided for the questionnaires.
    1. We are unsure of how to respond to this concern as each variable clearly reports that the tools are reliable and references to support these statements are included in the tex. Further clarity on this point would be helpful.
  9. Statistical Analysis: This is the primary concern. As highlighted in the introduction, various studies have explored these variables in isolation. This study would benefit from innovative insights, which could be achieved through multiple correlations or complex network models.
    1. Thank you for this comment, we have consulted a biostatistician and amended our analysis. These changes included amending all analysis to either generalised linear models or logistic regression. As such there have been extensive changes to the results section so we have not provided the entire results section in this response.
  10. This section lacks clarity and should not be repeatedly divided into subsections. The format resembles a report rather than a research article.
    1. This will no longer be relevant with the changes we have made.

Reviewer 3 Report

Comments and Suggestions for Authors

Thank you for your efforts on this work. Based on the review of this manuscript:

  1. The study makes minimal contribution to the field. The correlation between psychological factors like anxiety and depression with physical symptoms is well-documented. Unfortunately, this study provides little new theoretical insights or clinical significance. The results merely confirm existing knowledge. For instance, the correlation coefficients with pain (anxiety r=0.383, depression r=0.282) are only weak to moderate and add little to current literature. Due to the lack of group comparisons, this correlation might even be intrinsic to the patients themselves, considering the prevalence of anxiety and depression in modern populations.
  2. The small sample size (n=68) severely limits the statistical power and reliability of the findings. The use of online questionnaires introduced self-selection bias, and the geographic concentration of participants (50% Australian) further limits result generalizability. Recruitment methods primarily through social media platforms like globally-operated Facebook groups suggest potential sampling bias.
  3. The study conducts multiple correlation analyses without proper correction for multiple testing. Scatter plots in Figures 1-3 show weak relationships with high variability (presentation seems of little value). The authors claim "moderate correlations" with r-values below 0.4, which may be statistically significant but clinically irrelevant. Additionally, no power analysis was provided to justify the sample size.
  4. The Results section contains redundant information, and Figures 1-3 could be consolidated. The tables include excessive demographic details without clear relevance, and the Discussion section inadequately addresses the study's limitations.

Comments on the Quality of English Language

Written in acceptable English

Author Response

  1. The study makes minimal contribution to the field. The correlation between psychological factors like anxiety and depression with physical symptoms is well-documented. Unfortunately, this study provides little new theoretical insights or clinical significance. The results merely confirm existing knowledge. For instance, the correlation coefficients with pain (anxiety r=0.383, depression r=0.282) are only weak to moderate and add little to current literature. Due to the lack of group comparisons, this correlation might even be intrinsic to the patients themselves, considering the prevalence of anxiety and depression in modern populations.
    1. It is unfortunate that you feel this study provides minimal contribution. As the only study to have explored the association between psychological factors and tendinopathy severity (using a validated outcome measure) we feel these results are extremely novel.
    2. As the research team included several clinicians, and these data have been presented at international clinical conferences, we can confirm that the data are of definite interest to clinicians – in particular the use of a single question related to rupture.
    3. We are unsure why the reviewer feels that weak to moderate associations are a problem. As an example, the TENDINS-A has a -0.639 correlation with the VISA-A and the VISA-A is proposed to measure an identical construct to the TENDINS-A (disability). We feel that in a condition that has been proposed to be a predominantly peripheral pain condition that we should expect nothing more than weak to moderate associations between these conditions and that only encouraging publication of studies with larger correlations may introduce publication bias.
    4. Finally, in response to the query related to the correlation potentially being intrinsic, as per any cross-sectional study you are entirely correct that we are not able to report worse psychological status directly causes worse tendinopathy pain, symptoms and function. This would require a different study design and is why in our discussion section we never infer causality, and in our limitations section specifically state causality cannot be inferred from this research.

“Our study only explored associations, so we are unable to make any inference in relation to causality.”

  1. The small sample size (n=68) severely limits the statistical power and reliability of the findings. The use of online questionnaires introduced self-selection bias, and the geographic concentration of participants (50% Australian) further limits result generalizability. Recruitment methods primarily through social media platforms like globally-operated Facebook groups suggest potential sampling bias.
    1. We strongly disagree that a sample of 68 participants limits the statistical power and reliability of these findings. As the reviewer themselves has reported, this study was able to detect small correlations, with the majority of the results being reported as significant. If our sample was underpowered as the reviewer suggests, and at risk of Type II error then we would have issues with ‘False Negatives’, which does not appear to be a problem with our study. Furthermore, our study has a sample size larger than the majority of studies previously reported to evaluate psychological factors in tendinopathy (as per the Stubbs et al. 2020 review).
    2. We are unsure of the comment related to reliability as this study is cross-sectional and no reliability analyses were performed.
    3. We agree that a sample of predominantly Australian based participants, which is why this had already been included as a limitation in the discussion. We have included that suggestion to include social media as potentially introducing sampling bias, whilst recognising this was only one aspect of our recruitment strategy.

“The population in our study was predominantly Australian ethnicity, which may limit the generalisability of our results to other countries or ethnicities. Our recruitment method did include dissemination of our research flyer via social media, which may predispose to sampling bias towards participants with more disability or more severe psychological factors. However, given the large distribution of TENDINS-A scores, and the relatively low prevalence of severe psychological factors our data would suggest this is unlikely.”

  1. The study conducts multiple correlation analyses without proper correction for multiple testing. Scatter plots in Figures 1-3 show weak relationships with high variability (presentation seems of little value). The authors claim "moderate correlations" with r-values below 0.4, which may be statistically significant but clinically irrelevant. Additionally, no power analysis was provided to justify the sample size.
    1. Thank you for this comment, we have amended the analysis to perform more complex models and we feel this adequately addresses all the points raised.
  2. The Results section contains redundant information, and Figures 1-3 could be consolidated.
    1. Thank you for this comment. The figures have been removed as we now have an updated statistical analysis in the results.
  3. The tables include excessive demographic details without clear relevance.
    1. We have provided a further justification in the methods section as to why we feel it is vital that the included participants in sports medicine research are appropriately described.

“The broad reporting of demographic variables is designed to comply with international recommendations on improved reporting of who is included in sports medicine and tendinopathy research. This is important when many factors (e.g., ethnicity, sex or socioeconomic status) are poorly reported and/ or often linked to treatment outcomes. “

  1. The discussion section inadequately addresses the study's limitations.
    1. The only limitations to our study you have reported throughout your review is concern related to causality, geographical representation of the sample and the use of social media as a recruitment tool, which we have responded to and amended as above. We hope this is sufficient.